# LEARNING SURFACE PARAMETERIZATION FOR DOCUMENT IMAGE UNWARPING

## ABSTRACT

In this paper, we present a novel approach to learn texture mapping for a 3D surface and apply it to document image unwarping. We propose an efficient method to learn surface parameterization by learning a continuous bijective mapping between 3D surface positions and 2D texture-space coordinates. Our surface parameterization network can be conveniently plugged into a differentiable rendering pipeline and trained using multi-view images and rendering loss. Recent work on differentiable rendering techniques for implicit surfaces has shown high-quality 3D scene reconstruction and view synthesis results. However, these methods typically learn the appearance color as a function of the surface points and lack explicit surface parameterization. Thus they do not allow texture map extraction or texture editing. By introducing explicit surface parameterization and learning with a recent differentiable renderer for implicit surfaces, we demonstrate state-of-the-art document-unwarping via texture extraction. We show that our approach can reconstruct high-frequency textures for arbitrary document shapes in both synthetic and real scenarios. We also demonstrate the usefulness of our system by applying it to document texture editing.

## 1 INTRODUCTION

Reconstructing 3D shapes from images is a core problem in computer vision and graphics research. With the progress in differentiable rendering (Sitzmann et al., 2019b; Kato et al., 2018; Niemeyer et al., 2020; Li et al., 2018; Liu et al., 2019b), recent learning-based 3D reconstruction approaches have achieved impressive results using 2D supervision from single image (Chen & Zhang, 2019; Groueix et al., 2018; Choy et al., 2016; Mescheder et al., 2019; Wang et al., 2018) or multi-view images (Tang & Tan, 2018; Yariv et al., 2020). These methods achieve high quality 3D reconstruction using differentiable rendering with various 3D representations such as 3D mesh (Wang et al., 2018), volumetric representation (Mildenhall et al., 2020), or implicit functions (Mescheder et al., 2019). In recent neural rendering methods such as NeRF (Mildenhall et al., 2020) and IDR (Yariv et al., 2020), continuous representations such as volume or implicit functions achieve significantly better reconstruction results than meshes or voxels because they do not discretize the 3D surface a priori. However, these continuous representations usually do not encode explicit surface parameterization, allowing 3D shape re-texturing, editing the existing texture in the 2D texture space, or recovering 2D texture from 3D surfaces. One of the most direct applications of 2D texture recovery in a geometrically constrained manner, is document unwarping, i.e., inference of a document's flatbed-scanned version from a casual photo of a potentially creased document. Whereas 2D texture recovery could be equally valuable for other domains such as garments, or faces, the existing datasets are not directly applicable to our method.

Our novel texture mapping approach learns surface parameterization for document unwarping by learning continuous bijective functions between 3D surface positions and 2D texture-space coordinates. We use a signed distance function (SDF) (Chan & Zhu, 2005) to represent geometry and model the appearance as a function of the 2D texture coordinates. By utilizing implicit differentiable rendering (IDR), (Yariv et al., 2020) we can reconstruct 3D shape and learn the corresponding UV parameterization of the surface simultaneously using a per-pixel rendering loss and appropriate geometric regularizations.

We utilize two fully connected multi-layer perceptrons (MLPs) to learn a bijective mapping between 3D shapes and 2D texture space. More specifically, the *forward* MLP maps the 3D surface

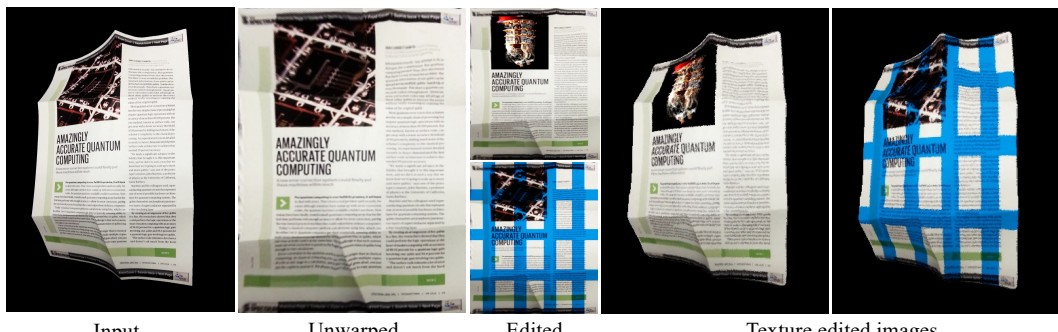

| Input | Unwarped | Edited | Texture edited images |

Figure 1: Proposed forward-backward network can be utilized in unwarping or editing the surface texture: The flattened texture can be edited and warped back to produce a texture edited image.

coordinates to 2D texture coordinates and the *backward* MLP maps the 2D texture coordinates to corresponding 3D surface coordinates. Following IDR (Yariv et al., 2020), we obtain the 3D surface coordinates by sphere-tracing along the ray, cast through each pixel. Our appearance rendering is formulated as a function of the 3D and the texture coordinates. Therefore, the forward and backward MLPs can be trained with a 2D pixel-wise loss between the rendered image and the given ground truth image. To the best of our knowledge, this is the first neural rendering method that can learn effective UV parameterization for implicit surfaces.

As a corollary, our method is also the first method which utilizes implicit surface based neural rendering for document unwarping. It is a challenging task due to the presence of geometric and photometric distortions in the document. For this particular problem we introduce a prior for shape-specific texture mapping to initialize the forward MLP (3D to 2D mapping). This prior is learned from a large dataset of UV mapped document meshes, assuming that document texture space maps to a 2D equiangular quadrilateral. This assumption regularizes the forward MLP to output a high-quality texture space that avoids degenerate solutions (see Fig. 3). Moreover, we introduce a conformality constraint in the backward MLP, which is consistent with how paper folds happen in the physical world, i.e., without any stretch or tear. This constraint also ensures that the backward function is bijective and smooth (Petrini et al., 2018).

The main contributions of our paper are the following: First, we propose an efficient way to learn texture parameterization for implicit neural representations using a differentiable rendering framework. Without 3D supervision, it only requires multi-view images as ground-truth and a texture mapping prior. Second, we show that our method can be effectively used for document unwarping tasks by learning a prior for explicit texture mapping on the document shape. We show that this prior can be learned from a dataset of texture-mapped meshes. Third, we show that our method is effective for document image unwarping and texture editing (see Fig. 1). We achieve a $52\%$ relative improvement over the publicly available state-of-the-art[1] (Das et al., 2019) in terms of mean local distortion across 500 views from ten synthetic scenes. Additionally, we achieve a $\sim25\%$ improvement in optical character recognition (OCR) in terms of character and word error rate.

## 2 PREVIOUS WORK

**Neural Rendering.** Neural rendering generates images and videos by integrating conventional computer graphics rendering pipelines into deep neural networks (Tewari et al., 2020). It enables explicit or implicit control of scene properties, including illumination, geometry, texture, *etc*. Neural rendering can synthesize semantic photos (Park et al., 2019b; Bau et al., 2019), novel views (Hedman et al., 2018; Sitzmann et al., 2019a), relighting (Xu et al., 2018; Meka et al., 2019), facial/body reenactment (Chan et al., 2019; Wei et al., 2019), estimate scene properties *etc*. Kato (Kato et al., 2018) proposed a differentiable neural renderer using an approximate gradient for rasterization. Liu (Liu et al., 2019a) proposed SoftRas, which extended differentiable rasterization. Li (Li et al., 2018) further demonstrated the feasibility of integrating ray-tracing in deep neural networks. More recently, implicit surface or volume rendering has become mainstream in neural rendering approaches such as IDR (Yariv et al., 2020) and NeRF (Mildenhall et al., 2020). These approaches are based

---

[1] A more recent approach, CREASE (Markovitz et al., 2020), data and models are not publicly available.

on multi-view surface reconstruction to associate the scene geometry to the appearance in different views. NeRF is extended to lot of variants including PixelNeRF (Yu et al., 2020), MVSNeRF (Chen et al., 2021), dynamic NeRF (Li et al., 2020; Pumarola et al., 2020), GRAF (Schwarz et al., 2020) and so on.

**Texture Mapping.** Texture mapping is an essential step in the computer graphics rendering pipeline. It defines a correspondence between a vertex on the 3D mesh and a pixel in the 2D texture image. To find such a mapping, FlexiStickers (Tzur & Tal, 2009) required users to specify a sparse set of correspondences. Bi  (Bi et al., 2017) proposed a patch-based texture mapping method using the 3D shape and images from multiple views. Morreale  (Morreale et al., 2021) used networks to represent 3D surfaces/shapes. Besides the above general texture mapping methods, some approaches focus on a specific object categories such as faces (Deng et al., 2018; Chen et al., 2019) and human bodies (Mir et al., 2020; Zhao et al., 2020). Recently, AtlasNet (Groueix et al., 2018) represented a 3D mesh as a collection of parametric surfaces; thus, texture mapping is trivial to obtain from a 2D parametric surface. A similar idea was adopted by Bednarik  (Bednarik et al., 2020) where they introduced geometric constraints when learning the decomposition. More recently NeuTex (Xiang et al., 2021) aims to recover the texture of a subject using NeRF (Mildenhall et al., 2020). However, NeuTex uses a spherical UV domain without any geometric constraints. Therefore, it is not suitable for document unwarping. Moreover, since NeRF (Mildenhall et al., 2020) doesn't learn an explicit geometry, NeuTex requires a coarse point-cloud to initialize the *backward* MLP. With an SDF based (Yariv et al., 2020) approach, our approach does not require such an initialization routine. We jointly learn the texture mapping and the geometry from scratch.

**Document Unwarping.** Document unwarping is a special application of texture mapping: the 3D object is usually a rectangular piecewise-developable surface, and the texture is well-structured, containing straight text lines, (usually) rectangular text blocks and figures, *etc*. Previous work usually adopted a two-step methodology: 1) 3D surface estimation and 2) deformed surface flattening. The 3D surface of a deformed document can be estimated from shading (Wada et al., 1997), multi-view images (Ulges et al., 2004), text lines (Tian & Narasimhan, 2011), local character orientations (Meng et al., 2018), document boundaries (Koo et al., 2009), and learning-based strategies (Pumarola et al., 2018). Flattening the obtained 3D surface always involves an expensive optimization process under certain geometry constraints such as conformality (You et al., 2017) or isometries (Bartoli et al., 2015). Flattening could be easier if the obtained 3D shape had a low dimensional parameterization like Generalized Cylindrical Surface (GCS) (Kil et al., 2017). Some studies (Das et al., 2017; Liang et al., 2008; Meng et al., 2015) proposed to unwarp each patch on the surface individually and then stitch the unwarped patches together. In recent years, data-driven methods (Ma et al., 2018; Das et al., 2019; Li et al., 2019; Markovitz et al., 2020; Das et al., 2021) have addressed document unwarping by leveraging large-scale synthetic datasets. These datasets contain deformed document images and their corresponding ground-truth UV coordinates. Methods trained on synthetic images often suffer from generalization performance due to the domain gap between synthetic and real data. In this paper, we utilize neural rendering techniques to learn a surface parameterization of a deformed document. We simultaneously estimate both 3D shapes and UV coordinates with a cycle consistency loss and geometric constraints. By leveraging the information from multi-view images, the proposed method demonstrates better document unwarping performance compared to a previous state-of-the-art, Das et al. (2019).

# 3 METHOD

A schematic diagram of the proposed approach is shown in Fig. 2. We utilize a recent differentiable rendering method, IDR (Yariv et al., 2020) for surface reconstruction and jointly learn the texture mapping of the learned implicit surface using two MLPs. In Sec. 3.1 we first describe some preliminaries about surface parameterization and IDR.

## 3.1 PRELIMINARIES

**Surface Parameterization.** The problem of surface parameterization focuses on finding a bijective mapping $F$ between a surface $Z \in \mathbb{R}^3$ and a polygonal domain $\Omega \in \mathbb{R}^n$. For a parametric or discrete surface representation, we can explicitly compute this mapping (Tzur & Tal, 2009) using constrained optimization. In contrast, implicit surfaces are represented as continuous functions and cannot be readily parameterized. In this paper, we propose to learn such bijective mapping between a learned

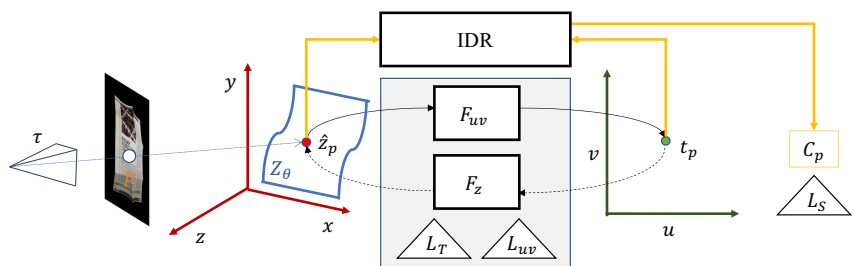

Figure 2: Proposed surface parameterization learning using the forward ($F_{uv}$) and backward MLP ($F_z$): Given camera pose $\tau$, and a pixel $p$ we jointly learn the geometry represented by a SDF $Z_\theta$, the $F_{uv}$, and the $F_z$. $\hat{z}_p$ is the ray-surface intersection point in 3D domain and $t_p$ is the corresponding texture coordinate in UV domain. The yellow arrows denote the input and output of the IDR (Yariv et al., 2020), and $C_p$ is the predicted RGB color. Triangles denote the losses defined in Eq. 12.

implicit surface and a 2D planar domain $\Omega \in \mathbb{R}^2$ using our proposed forward and backward MLPs. $\Omega$ is the texture space or UV space, parameterized using 2D UV coordinates $\mathbf{t} = (u, v)$. We can use any continuous parameterization function as the UV space. Since this work particularly focuses on document unwarping, we choose the UV space to be a regular 2D grid.

**Implicit Differentiable Rendering.** Implicit Differentiable Rendering (Yariv et al., 2020) reconstructs the geometry of an object from multi-view images as the zero level set, $Z_\theta$ of an MLP $S$,

$$Z_\theta = \{\mathbf{z} \in \mathbb{R}^3 \mid S(\mathbf{z}; \theta) = 0\} \tag{1}$$

where $\theta$ are the learnable parameters. To render the surface $Z_\theta$, IDR uses another MLP to model the radiance (RGB color) as a function of the surface point ($\mathbf{z}_p$), corresponding surface normal ($\mathbf{n}_p$), view direction ($\mathbf{v}_p$) and a global geometry feature vector ($\mathbf{g}_p$):

$$C_p = A(\mathbf{z}_p, \mathbf{n}_p, \mathbf{v}_p, \mathbf{g}_p) \tag{2}$$

Here, $C_p$ denote the predicted color at pixel $p$ and $A$ denotes the appearance MLP. The surface point is obtained by a sphere-tracing method (Hart, 1996) along the ray $r_p(\tau)$ through pixel $p$. $\tau \in \mathbb{R}^k$ denotes camera parameters of the scene. Additionally, IDR also presents a differentiable way to obtain a ray and geometry intersection point ($\hat{\mathbf{z}}_p$) as a function of the camera ray. Although, the IDR can disentangle geometry and appearance, it only allows to re-texture a new geometry with a learned appearance MLP, $A$. Editing a texture or extracting a surface texture map is not possible in a vanilla IDR framework since no explicit texture mapping is learned.

## 3.2 LEARNING SURFACE PARAMETERIZATION

To learn a meaningful parameterization of the implicit surface $Z_\theta$, we represent the radiance at pixel $p$ as a function of the UV space. To this end, we modify the IDR model (Eq. 2):

$$C_p = A_{uv}(\mathbf{t}_p, \mathbf{z}_p, \mathbf{n}_p, \mathbf{v}_p, \mathbf{g}_p) \tag{3}$$

The texture parameterized appearance MLP is modeled as a function of the texture coordinate $\mathbf{t}_p$ at surface point $\mathbf{z}_p$, corresponding to a pixel $p$. We can jointly train the surface MLP ($S$) and texture parameterized appearance MLP ($A_{uv}$) using a pixel wise rendering loss between the predicted radiance ($C_p$) and ground-truth radiance ($C_p^{gt}$) at pixel $p$.

**Forward and backward texture parameterization.** We represent the mapping between the 3D surface and 2D texture space using the *forward* function $F_{uv}$:

$$\mathbf{z} \to \mathbf{t}. \tag{4}$$

The $F_{uv}$ is modeled as an MLP. It is trained by mapping a ray-surface intersection point $\hat{\mathbf{z}}_p$ to its corresponding texture coordinate $\mathbf{t}_p$ corresponding to a pixel $p$. Now to establish the bijective mapping (discussed in Sec. 3.1) between the surface and texture space we utilize a *backward* function $F_z$:

$$\mathbf{t} \to \mathbf{z}. \tag{5}$$

$F_z$ is an MLP that learns an inverse mapping between the texture and the 3D space. It is trained by mapping a texture coordinate $\mathbf{t}_p$ to its corresponding ray-surface intersection point $\hat{\mathbf{z}}_p$.

**Shape specific prior for $F_{uv}$.** Jointly training the forward, backward and rendering network leads

to the wrong UV mapping with local minima (see Fig. 3) where multiple $\hat{\mathbf{z}}_p$ map to a single texture coordinate. To avoid such degenerate cases, we initialize $F_{uv}$ with a texture mapping prior, learned from a large dataset of UV mapped meshes. This learned prior ($\hat{F}_{uv}$) makes the learned texture mapping suitable for document unwarping. We assume the document shape to be a deformed quadrilateral and the corresponding UV space to be a regular grid ($\in [0.0, 1.0]$). The top leftmost and the bottom rightmost 3D coordinate of the shape maps to $(u, v) = (0, 0)$ and $(u, v) = (1, 1)$ respectively. To learn $\hat{F}_{uv}$ we utilize a collection of UV mapped document meshes from the Doc3D (Das et al., 2019) dataset and train an MLP with the same parameters as $F_{uv}$. For each scene, we use $\hat{F}_{uv}$ to initialize the weights of $F_{uv}$ and train jointly with $S$ and $A_{uv}$.

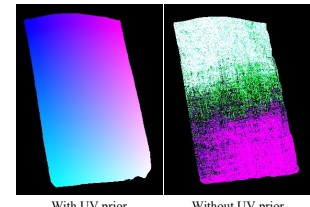

With UV prior Without UV prior

Figure 3: Without a prior the forward network, $F_{uv}$ leads to degenerate cases: multiple 3D points, $\hat{z}_p$ are mapped to the same texture coordinate $t_p$.

**Deformation constraints for $F_z$.** Conformal map (Haker et al., 2000) allows a 3D domain to be mapped to a texture domain with low distortion satisfying the bijective property between domains. We use a conformality constraint for $F_z$ to ensure the deformation properties mentioned above. We define the conformality constraint in terms of the metric tensor, $\mathbf{J}^\top \mathbf{J}$ of the $F_{\mathbf{z}}$, where $\mathbf{J}$ is the jacobian of $F_{\mathbf{z}}$ (Eq. 6):

$$\mathbf{J} = \begin{bmatrix} \dfrac{\delta F_{\mathbf{z}}}{\delta u} & \dfrac{\delta F_{\mathbf{z}}}{\delta v} \end{bmatrix} = [D_u \ D_v] \qquad \mathbf{J}^\top \mathbf{J} = \begin{bmatrix} D_u^\top D_u & D_u^\top D_v \\ D_u^\top D_v & D_v^\top D_v \end{bmatrix} = \begin{bmatrix} E & F \\ F & G \end{bmatrix} \quad (6)$$

The conformality constraint is defined as $\mathbf{J}^\top \mathbf{J} = \beta \mathbf{I}$. Here $\beta$ is a unknown local scaling function and $\mathbf{I}$ is the identity matrix. For developable surfaces which can be physically flattened without any stretch e.g. papers, $\beta$ doesn't vary across the parameterization space. Therefore, we consider a fixed global scale ($[\beta_u^g, \beta_v^g]$) for the conformality constraint.

**Unwarping by sampling $F_{\mathbf{z}}$.** To unwarp an input image, we determine the pixel at $p = (x, y)$ in the input image should be projected to $(u, v)$ in the unwarped image. Here the unwarped image refers to the texture space. The coordinates $(u, v)$ and $p$ are associated by $F_{\mathbf{z}}$ and $\tau$: For a $(u, v)$ coordinate, its corresponding point in 3D is obtained by $\hat{z}'_p = F_{\mathbf{z}}(u, v)$. Given the camera parameter $\tau$, $\hat{z}'_p$ is projected to $p$ in the input image. Thus for each pixel in the unwarped image, we can find its corresponding pixel in the input image which is all we need for unwarping.

### 3.3 Loss Functions

We use the rendering losses on the predicted color, $C_p$, and predicted document mask $M_p$ at pixel $p$ to train the geometry $S$. Here $M_p \in \{0, 1\}$ refers to whether the pixel $p$ is occupied ($M_p = 1$) by the shape or not ($M_p = 0$). We assume masks are provided as input. Additionally, we employ appropriate regularization losses to jointly train $S$, $A_{uv}$, $F_{uv}$ and $F_z$.

**Loss for $S$.** Following IDR (Yariv et al., 2020), for each $p$ we apply a sphere tracing (Hart, 1996) algorithm to find the intersection point of the ray $r_p(\tau)$ and the surface $Z_\theta$. Given the ground-truth RGB color $C_p^{gt}$ and the predicted RGB color $C_p$, the RGB loss is defined as:

$$L_{rgb} = \frac{1}{|P|} \sum_{p \in P_{in}} \left\| C_p^{gt} - C_p \right\|_1 \quad (7)$$

Where $P$ is the set of pixels in the minibatch. The pixels $P_{in} \subset P$ for which ray surface intersection has been found and $M_p = 1$. The mask loss is defined as:

$$L_{mask} = \frac{1}{\alpha |P|} \sum_{p \in P_{out}} CE(M_p^{gt}, M_p) \quad (8)$$

Here $P_{out} = P \setminus P_{in}$, alpha is a tunable parameter and $CE(.)$ is the cross-entropy loss. The value of $M_p = \mathcal{M}_{p,\alpha}(\theta, \tau)$ is a differentiable function of the learned $Z_\theta$ (Yariv et al., 2020). Additionally, to force $Z_\theta$ to be a approximate signed distance function we use Eikonal Regularization (Gropp et al.,

2020):

$$L_{ek} = \mathbb{E}_z (\|\nabla_z S(\mathbf{z}; \theta)\| - 1)^2 \tag{9}$$

where $z$ denotes uniformly sampled points within a bounding box of the 3D domain.

**Loss for $F_{\mathbf{uv}}$.** Although we initialize $F_{\mathbf{uv}}$ with learned prior parameters, we constrain the predicted 2D texture coordinates during training in order to avoid non-uniform mapping of the 3D and the UV domain which can squeeze or stretch the warped texture (example in supplementary). We employ a Chamfer distance between the $\mathbf{t}_p$ and uniformly sampled 2D points $\mathcal{T} \in [0, 1]$ to ensure $F_{\mathbf{uv}}$ approximately outputs $\mathcal{U} \sim [0, 1]$. This regularization term is defined as:

$$L_{uv} = CD_{p \in P_{in}}(\mathcal{T}, \mathbf{t}_p) \tag{10}$$

here $CD(.)$ denotes the Chamfer distance and $t_p$ the predicted texture coordinates corresponding to ray-surface intersection points $\hat{\mathbf{z}}_p$.

**Loss for $F_{\mathbf{z}}$.** $\hat{z}'_p$ is the output of $F_{\mathbf{z}}$. $F_{\mathbf{z}}$ is trained with weighted regression loss between $\hat{z}_p$ and $\hat{z}'_p$:

$$L_z = \frac{1}{|P_{in}|} \sum_{p \in P_{in}} w_p (\hat{z}_p - \hat{z}'_p)^2 \tag{11}$$

$w_p$ is a pre-calculated per-pixel weight based on the document mask ($M$) which assigns higher value to the pixels at the boundary of the document. (More weight calculation details in Supplementary).

Additionally, to constrain $F_{\mathbf{z}}$ to be a fixed scale conformal mapping (Bednarik et al., 2020). We employ three constraints on the elements of the metric tensor $E$, $F$ and $G$ defined in Eq. 6.

$$L_E = \frac{1}{|P_{in}|} \sum_{p \in P_{in}} (E_p - \tilde{E})^2 \quad L_G = \frac{1}{|P_{in}|} \sum_{p \in P_{in}} (G_p - \tilde{G})^2 \quad L_F = \frac{1}{|P_{in}|} \sum_{p \in P_{in}} (F_p)^2$$

Here $\tilde{E}$ and $\tilde{G}$ is the mean of $E$ and $G$.

Our combined loss function is defined as:

$$L = \underbrace{(L_{rgb} + \gamma_1 L_{mask} + \gamma_2 L_{ek})}_{L_S} + \rho L_{uv} + \underbrace{(\delta_1 L_z + \delta_2 L_E + \delta_3 L_G + \delta_4 L_F)}_{L_T} \tag{12}$$

Here $\gamma$, $\rho$ and $\delta$ denote the hyperparameters associated with the losses.

### 3.4 TRAINING DETAILS

The surface MLP $S(\mathbf{z}, \theta)$ consists of 8 layers with a hidden layer dimension of 128, with a skip connection to the middle layer (Park et al., 2019a). Following IDR (Yariv et al., 2020), $S$ is initialized to produce an approximate SDF of a unit sphere. The rendering network $A_{uv}$ has 4 layers with hidden layer dimension of 512 and uses a sine activation function (Sitzmann et al., 2020) at each layer. $F_{uv}$ and $F_z$ share identical architecture with 8 layers with 512 dimensional hidden units and sine activation (Sitzmann et al., 2020). Following NeRF (Mildenhall et al., 2020), we use a $k$ dimensional Fourier mapping ($\chi_k : \mathbb{R} \to \mathbb{R}^{2k}$) to learn high frequency details in the shape, RGB and the UV space. For $S$, $A_{uv}$ we follow the setting of (Yariv et al., 2020), and set $k = 6$ and $k = 4$ respectively. For $F_{uv}$ and $F_z$ we empirically set number of Fourier bands $k = 10$. We start with an initial learning rate of $1e \times^{-5}$ and train for 150K iterations by halving the learning rate after every 50K iterations. Initially, $\alpha$ is set to 50 and doubled during the training after every 50K iterations. We set $\gamma_1 = 100.0$, $\gamma_2 = 0.1$ and $\rho = 0.001$. $\delta_1$ is set to 0.001 for the initial 30K iterations. Afterward, $\delta_1$ is multiplied by a factor 2 at every 10K iterations for a maximum of 7 times. $\delta_2$, $\delta_3$ and $\delta_4$, are set to zero for the initial 100K iterations. Only $L_z$ is sufficient to achieve a good texture to 3D mapping during the shape optimization phase. Afterwards we set $\delta_2 = \delta_3 = 0.001$ and $\delta_4 = 0.01$. The metric tensor calculation is implemented using auto-differentiation.

## 4 EXPERIMENTAL RESULTS

First, we quantitatively compare the proposed method with state-of-the-art document unwarping method DewarpNet (Das et al., 2019). Our quantitative and qualitative experiments are performed on 10 synthetic scenes and 10 real scenes. Second, we apply our method to texture editing. Last, we conduct ablation studies to demonstrate the effectiveness of our proposed loss functions.

## 4.1 EVALUATION DATASET AND METRICS

Our synthetic evaluation data consists of 10 scenes rendered using Blender following a rendering pipeline similar to Doc3D. Each scene consists of 50 random views sampled from a $45^o$ solid angle in the upper hemisphere. The real-world evaluation data consists of 3 scenes from the dataset of (You et al., 2017), and 9 scenes captured by us. Each scene consists of 5-20 images per scene. We manually annotate the masks for each scene. To obtain camera poses for the real-world data, we utilize the COLMAP (Schönberger & Frahm, 2016) multi-view reconstruction pipeline. Both synthetic and real data include the document scan, as the unwarping ground-truth.

We use image-based evaluation metrics for quantitative evaluation, including Local Distortion (LD) and Multi-Scale Structural Similarity (MS-SSIM). These are standard metrics used for document unwarping evaluation (Das et al., 2019; Ma et al., 2018). LD is based on dense SIFT flow (Liu et al., 2011) between the unwarped and scanned images. Image similarity metric, MS-SSIM (Wang et al., 2003) is based on local image statistics (mean and variance) of the unwarped and scanned (ground-truth) images calculated over multiple Gaussian pyramid scales. We use the same settings as (Das et al., 2019; Ma et al., 2018) for fair comparison.

## 4.2 DOCUMENT UNWARPING

The primary application of our learned forward and backward MLP is document unwarping. The quantitative comparison with the state-of-the-art model (Das et al., 2019) is shown in Table 1 for the synthetic and real scenes. In terms of average performance of all the views (*all views* col. in Table 1) we improve the LD by $\sim 52\%$ compared to (Das et al., 2019). Since we use multi-view images for training, our results are more consistent across all the views compared to DewarpNet, which is also a key reason for the significant improvement. We conjecture that (Das et al., 2019) as a single image unwarping method should perform well on simpler deformations and frontal view images. However, it is not always the case. In qualitative comparisons in Fig. 4, DewarpNet often generates artifacts even for reasonably frontal views and simple deformations. Comparatively, our results are qualitatively superior.

| Scene | DewarpNet (*all views*) | | DewarpNet (*best view*) | | Proposed (*all views*) | |
|---|---|---|---|---|---|---|
| | MSSIM ↑ | LD ↓ | MSSIM ↑ | LD ↓ | MSSIM ↑ | LD ↓ |
| Synth 1 | 0.42 | 9.54 | 0.68 | 3.29 | **0.74** | **2.59** |
| Synth 2 | 0.75 | 5.68 | **0.83** | **2.59** | 0.76 | 4.40 |
| Synth 3 | 0.73 | 7.80 | **0.85** | **2.94** | 0.78 | 5.44 |
| Synth 4 | 0.59 | 6.88 | 0.63 | **2.53** | **0.64** | 2.85 |
| Synth 5 | 0.48 | 7.11 | **0.64** | **3.13** | 0.61 | 4.55 |
| Synth 6 | 0.50 | 6.34 | **0.62** | **2.53** | 0.47 | 3.92 |
| Synth 7 | 0.52 | 7.99 | **0.76** | 2.64 | 0.74 | **2.55** |
| Synth 8 | 0.56 | 10.05 | **0.70** | **3.44** | 0.64 | 5.31 |
| Synth 9 | 0.49 | 7.48 | 0.73 | 1.87 | **0.78** | **1.56** |
| Synth 10 | 0.52 | 8.07 | **0.78** | **2.78** | 0.73 | 3.13 |
| Syn. Mean | 0.56 | 7.69 | **0.70** | **2.82** | 0.69 | 3.63 |
| Real 1 | 0.26 | 9.77 | **0.39** | 5.78 | 0.37 | **5.68** |
| Real 12 | 0.24 | 12.94 | 0.24 | 10.98 | **0.35** | **8.38** |
| Real 6 | 0.44 | 9.15 | **0.48** | **7.78** | 0.37 | 16.80 |
| Real Mean | 0.31 | 10.62 | **0.37** | **8.18** | 0.36 | 10.28 |

Table 1: Comparison with (Das et al., 2019) on synthetic scenes: *all views* refers to the mean error metric on all scene images, *best view* refers to the lowest possible error from an image in a scene.

We also report in a stricter evaluation scenario (*best view* column of Table 1) where we compare our results with the best possible numerical results achieved by DewarpNet from a single view in a scene. We perform better than DewarpNet in 91.2% of all views, however when the best view can be selected our method do slightly worse in 7 scenes. This 'stricter' setting shows quantitatively competitive results compared to DewarpNet with a oracle (practically challenging) view selector.The choice of the best unwarped result is often subjective. For a more comprehensive comparison, we qualitatively compare the best results of DewarpNet with our results across 6 scenes in Fig. 5. These 6 scenes are chosen among the 7 scenes for which DewarpNet **achieves a better quantitative result** than the proposed approach for at least one view. In Fig. 5 our results are clearly better than the DewarpNet in all cases, with straighter lines and better rectified structure. The evaluation scores do not accurately reflect the improvement due to the sensitivity of LD and MSSIM to subtle perceptually unimportant global transformations, such as translation of the image by few pixels. However, such transformations do not affect the visual quality or readability of the unwarped results. More discussion and qualitative comparison is available in supplementary material.

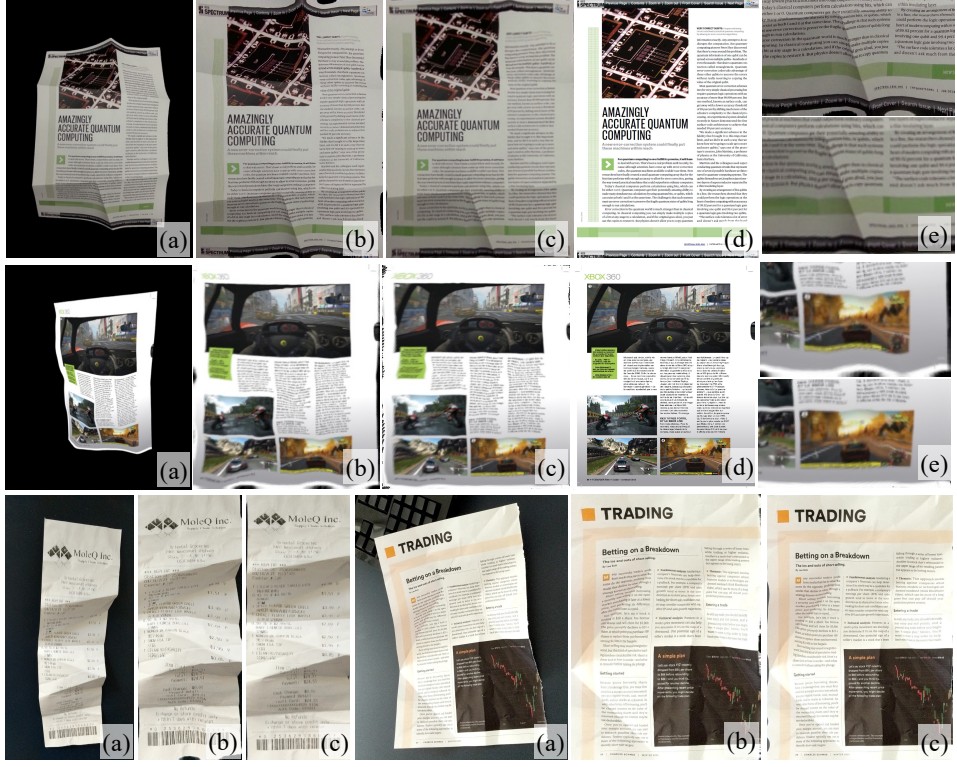

Figure 4: Qualitative comparison with DewarpNet (Das et al., 2019): (a) Input image, (b) Dewarp-Net unwarping, (c) proposed unwarping, (d) GT scanned image, (e) enlarged regions: DewarpNet (*top*), and proposed (*bottom*). We use reasonable frontal view of the document for a fair comparison.

| | DewarpNet | | | Proposed | | |
|---|---|---|---|---|---|---|
| | ED $\downarrow$ | CER ($std$) $\downarrow$ | WER ($std$) $\downarrow$ | ED $\downarrow$ | CER ($std$) $\downarrow$ | WER ($std$) $\downarrow$ |
| Mean | 798.30 | 0.2827 (0.12) | 0.4646 (0.17) | **600.78** | **0.2122** (0.10) | **0.3568** (0.11) |

Table 2: Comparison of OCR error metrics: We improve the OCR performance of Das et al. (2019) by ∼25% in terms of Edit Distance (ED), Character Error Rate (CER), and Word Error Rate (WER).

The quantitative comparison for real scenes are reported in Table 1 (bottom). We achieve better results in terms of mean and best evaluation score than DewarpNet in 2 out of 3 scenes. We notice that the evaluation results are a little worse for the real scenes than synthetic scenes due to the fewer available views (5-10 compared to 50). Moreover, there are cases like Real 6, which do not have sufficient texture. Such data are a failure case of IDR since there is insufficient information to reconstruct the 3D shape. As a result of the poor 3D shape, our texture parameterization network produces an inferior unwarping result (More details are available in Supplementary). We also report qualitative comparisons with You et al. (2017) and Das et al. (2019) on additional real documents in supplementary.

**OCR Evaluation.** We also evaluated the OCR performance on 5 real scenes across 77 images in Table 2. We use Edit Distance (ED) (Miller et al., 2009), Character Error Rate (CER) and Word Error Rate (WER) as our evaluation metrics. ED is defined as the total number of substitutions (s), insertions (i) and deletions (d) required to obtain the reference text, given the recognized text. The reference text is obtained by running the OCR algorithm on the scanned ground-truth image of each document. CER is defined as: $(s + i + d)/N$ where $N$ is the number of characters in the reference text. We use Tesseract 4.1.1 based LSTM OCR engine for this experiment. Our unwarped results reduce the ED, CER and WER by ∼25%. This improvement proves our unwarped results are more suitable for downstream applications tasks like OCR.

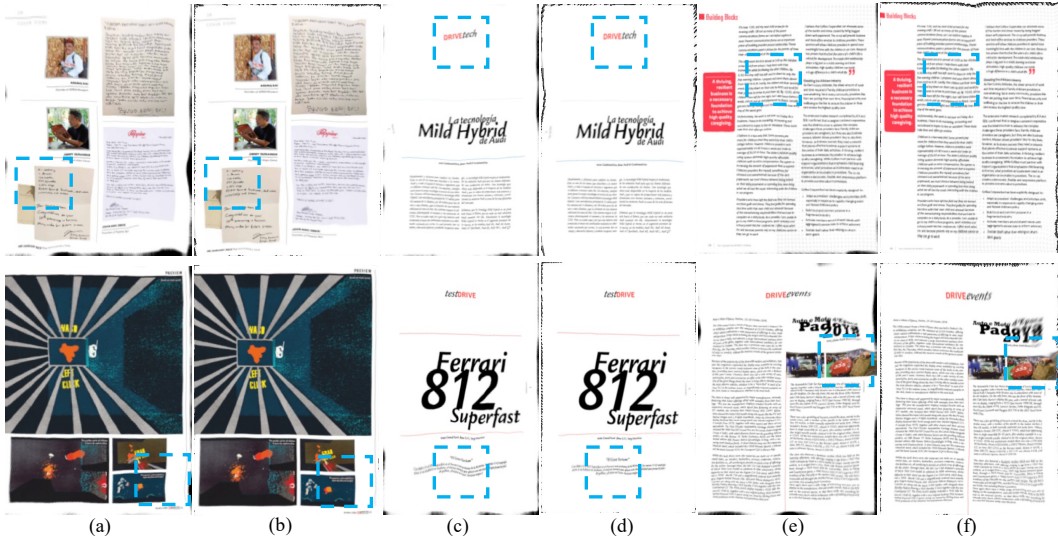

Figure 5: Comparison of DewarpNet (a,c,e) with the proposed unwarped result (b,d,f) for the view that yields the best LD with DewarpNet. Proposed results are clearly better, however this improvement is not captured by LD. Follow the blue dashed boxes for discrimitative regions.

**Texture Editing.** In addition to document unwarping, our proposed forward and backward MLP can also be used for high quality texture editing. We show two texture editing examples in Fig. 1. We use the backward MLP to unwarp the texture from the input image, then we edit the texture and warp it back to image space using the learned forward MLP. (More details in Supplementary).

**Ablation Study.** We ablate how loss terms $L_z$, $L_E$, $L_F$, and $L_G$ affect the unwarping results. We train $F_{uv}$ and $F_z$ with different combinations of these loss terms and report the mean MSSIM and LD in Table 3 (appendix). Qualitative results for one scene are shown in Fig. 6 (appendix).

## 5 TRAINING TIME, GENERALIZABILITY AND FUTURE WORK

Our proposed method for a scene can be trained in approximately 18 hours for $448 \times 448$ resolution images using a single Titan Xp GPU. The current training time per scene is very high compared to DewarpNet's inference time which makes it unsuitable for real time applications. However, this is a fast growing field and there are multiple other works that are focusing on improving the speed and generalization abilities (Garbin et al., 2021; Bergman et al., 2021) of neural rendering. Therefore, obtaining a faster training scheme is considered as a future work.

Our method can be applied to fabrics, which are very similar to papers and lead to practical applications of texture editing. However, none of the current 3D garment/fabric datasets (Patel et al., 2020) can be easily adapted to train the $\hat{F}_{uv}$ prior. For more complex UV spaces (e.g., texture atlas), learning the prior may require decomposing the shape to multiple simple UV maps. The proper way to do this is beyond the scope of this paper, however we believe it's an exciting future work. As importantly, in this paper, we have introduced a number of domain specific strong constraints that suit the rectangular paper shape. These constraints improve empirical results. More general objects will require different constraints e.g., spherical UV domain, local scaling of the conformal map etc.

## 6 CONCLUSIONS

We have introduced an end-to-end trainable architecture that can simultaneously learn texture parameterized 3D shapes from multi-view images. This is the first work to learn surface parameterization of an implicit neural representation to the best of our knowledge. We have demonstrated the applicability of our approach on multiple synthetic and real scenes for the task of document unwarping and document texture editing. We want to extend this method to learn surface parameterization for more complex shapes such as faces or general 3D objects in future work.

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

| Mean | b | b+w | b+c | b+w+c |
|---|---|---|---|---|
| LD $\downarrow$ | 11.2 | 10.50 | 6.24 | **3.63** |
| MSSIM $\uparrow$ | 0.4632 | 0.4622 | 0.5556 | **0.6888** |

Table 3: Weighted $L_z$, and conformality effects: $b$ is for the model trained without conformality constraints and with $w_p = 1$; w is for weighted $L_z$ and c is for the use of conformality constraints.

## 7 ETHICS STATEMENT

Texture editing application of our proposed approach can have both positive and negative societal impact. On the positive side, real document images can be gracefully redacted to protect sensitive information. On the contrary, it can be potentially used for editing real documents and change the content to commit fraud and spread misinformation.

## 8 REPRODUCIBILITY STATEMENT

We believe our results are reproducible by following the training details in Section 3.4 of main submission and Section 2, 3, 4, 5 from the supplementary material.

## A APPENDIX

### A.1 ABLATION STUDY

We ablate how loss terms $L_z$, $L_E$, $L_F$, and $L_G$ affect the unwarping results. We train $F_uv$ and $F_z$ with different combinations of these loss terms and report the mean MSSIM and LD in Table 3. Qualitative results for one scene are shown in Fig. 6 (appendix). In the basic version (listed as $b$), no conformality constraints ($L_E$, $L_F$ and $L_G$) are used and $w_p = 1$ in $L_z$. The $b+w$ version introduces a weighting function that assigns a higher value to $w_p$ if a pixel is closer to the document boundary. Introducing this loss improves the boundary; notice the white margin at the top and bottom in the second column of Fig. 6. Introducing the conformality constraints alleviates the unusual stretch in the texture and improves smoothness (Fig. 6, col. 3). Using both improves the boundary and the texture smoothness (Fig. 6, col. 4) and achieves the best result.

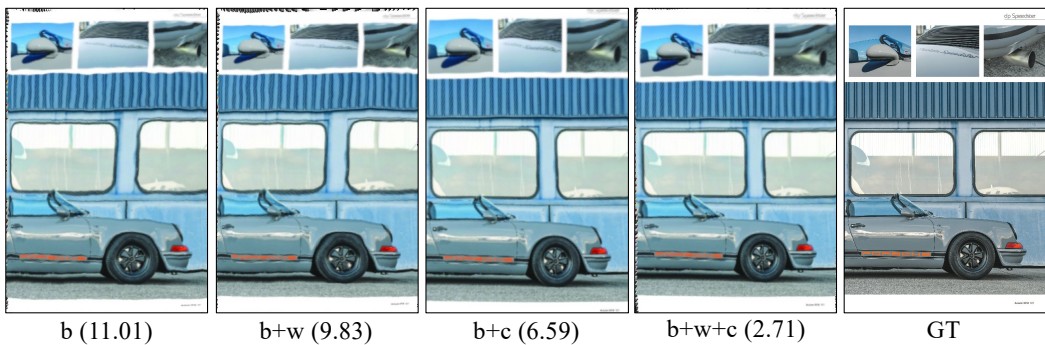

| b (11.01) | b+w (9.83) | b+c (6.59) | b+w+c (2.71) | GT |
|---|---|---|---|---|

Figure 6: Illustration of weighted $L_z$, and conformality effects: $b$ is for the model trained without conformality constraints and with $w_p = 1$; w is for weighted $L_z$ and c is for the use of conformality constraints. GT is the ground-truth scan. The number in parenthesis denote the respective LD values.

