# OpenReview forum: "Learning Surface Parameterization for Document Image Unwarping"
_ICLR.cc/2022/Conference — ICLR 2022 Submitted_

### Official Review · Reviewer_Cc2n · 2021-10-16

**Correctness:** 3
**Technical Novelty And Significance:** 3
**Empirical Novelty And Significance:** 3
**Recommendation:** 5
**Confidence:** 4

**Main Review:**

The method seems to be primarily based on IDR with following similarities:
- 3D surface is represented by a Signed Distance Field (SDF)
- Appearance is learned from a neural network that maps 3D surface geometry (location and normal) and viewing angle to radiance values
- The networks are trained with an overall loss including an RGBA rendering term and an Eikonal term for regularising SDF

IDR can handle specular materials but at the cost of introducing so-called 'shape-radiance' ambiguities. This means geometry cannot be possibly recovered due to view-dependency and is only regularised by neural nets' inherent priors (c.f. NeRF++). In this paper though, it seems document pages are all diffuse and I think it would make more sense to drop the view direction term $\mathbf{v}_p$ from Eq. 3. This would disambiguate geometry and could improve robustness.

The authors propose a forward-backward network to learn the square parameterisation of surface with a conformal loss on the forward network. While this design encourages bijective mapping through minimising $L_z$, there is no guarantee the resulting networks are truly bijective. Why not use structures that guarantee invertibility (e.g. Invertible Residual Networks, Behrmann et al.)?

In paragraph "Deformation constraints for Fz" the author wrote $\mathbf{J}^\top\mathbf{J}=\beta I$. Shouldn't this be $\mathbf{J}^\top\mathbf{J}=diag(\alpha,\beta)$ since you don't know the aspect ratio of document (which is what authors are actually doing with $L_E$,$L_G$)?

The results show clear improvments over existing methods and look very promising. Though personally I'm not familiar with the specific topic of document unfolding or methods that are compared against.





**Summary Of The Paper:**

The paper presents a multi-view method for undistorting documents. The method makes use of an existing method (IDR) for reconstructing distorted documents in 3-D, with an additional bijective neural network to parameterize distorted surfaces by rectangles. The method significantly outperforms SOTAs in the experiments.



**Summary Of The Review:**

I like the overall idea of this paper and the results look really convincing. However I would not recommend acceptance to ICLR given its significant overlap with an existing technique (IDR).

---

> ### Author Response · Authors · 2021-11-19
> **Response to Reviewer Cc2n**
>
> We thank the reviewer for the insightful comments. We address the concerns below:
>
> **Modeling without view directions:**
> Not all papers are perfectly diffuse, such as the glossy magazine pages (See figure 13 of supplementary). Therefore, removing the view direction from the renderer often results in bad 3D reconstruction leading to inferior unwarping results.
>
> **Use of invertible networks:**
> Employing invertible architectures is a very interesting future research objective. Invertible networks such as i-ResNet or GLOW [2] only model a single distribution. On the other hand, there are invertible flow models such as C-Flow which allow conditional modeling of two distributions, therefore allowing domain transfer. Such conditional models are suitable for our task (3D to UV). However, the training time of these networks is much slower than MLPs, the C-Flow model takes 10 days to be trained on 4 GPUs [1].
>
> **Conformality equation:**
> Mathematically, $J^{T}J=\beta I$ is the definition of isometry. To be more exact, the alternative equation suggested by is correct. We will clarify this part in the updated version of the paper.
>
> **Difference with IDR:**
> IDR is only utilized in our system for geometry reconstruction and it does not allow texture unwarping. We have introduced multiple novel components and modifications to enable texture unwarping. For example, our main contribution is the novel forward and backward MLPs to facilitate texture mapping.  Moreover, we introduce the UV prior and geometric constraints which we show are essential to training the proposed method. All of our contributions are novel with the formulation of unwarping based on a neural rendering based framework.
>
> [1] Pumarola, Albert, et al. "C-flow: Conditional generative flow models for images and 3d point clouds." Proceedings of the IEEE/CVF Conference on Computer Vision and Pattern Recognition. 2020
> [2] Kingma, Diederik P., and Prafulla Dhariwal. "Glow: Generative flow with invertible 1x1 convolutions." arXiv preprint arXiv:1807.03039 (2018).

---

> > ### Comment · Reviewer_Cc2n · 2021-12-02
> > **Some concerns remain**
> >
> > I'd like to thank authors for their response, and for providing additional results. I stand by my initial review that this is an interesting paper and the results are compelling. However, some concerns remain:
> >
> > - I still think IDR takes on a significant portion of solution, i.e. reconstructing documents in 3D.  The true contribution of this paper seems to be an forward-backward MLP-based surface parameterization method for texture mapping/unwarping. While this idea is indeed interesting, it should be noted that it has been explored albeit in a different task that slightly predates this paper [1].
> >
> > - Lambertian model for documents: I don't see any specular highlights in figure 13. While I agree that some real world documents could be glossy, the Lambertian assumption is sensible at least from the results presented in this paper, and would greatly simplify the appearance model and open other possibilities, e.g. whether it is possible to recover surface albedo independent from shading/lighting effects?
> >
> > [1] NeuTex: Neural Texture Mapping for Volumetric Neural Rendering, Xiang etal, CVPR 2021

---

> > > ### Author Response · Authors · 2021-12-02
> > > **Further clarification**
> > >
> > > We thank the reviewer for the comments. Following are the clarifications regarding the remaining concerns:
> > > * We have discussed the contributions of NeuTex in the related work and would like to highlight that it doesn't allow unwarping documents due to the use of spherical UV domain without a shape specific prior in the forward MLP and the missing geometric constraints in backward MLP. Consequently, the learned backward mapping is not as good as the proposed approach.
> > >
> > > * Figure 13 doesn't show specular highlights but presents a type of paper that is somewhat glossy (notice the intensity changes on the black figure region). We agree that simplifying the appearance model to Lambertian will allow disentangling the reflectance and shading. However, we believe a more complicated training (more regularizations) and specific data (e.g. shading free images) will be required to accurately model each component.

---

### Official Review · Reviewer_hfPz · 2021-11-01

**Correctness:** 3
**Technical Novelty And Significance:** 3
**Empirical Novelty And Significance:** 3
**Recommendation:** 6
**Confidence:** 4

**Main Review:**

Strengths:
This paper seems to be the first paper where implicit differentiable rendering is applied to document unwrapping.

The proposed method is reported to be par with the baseline method (DewrapNet) in quantitative comparisons and outperformed the baseline in qualitative comparisons.

Weakness:
The proposed method is similar to NERF, where MLPs are trained for each subject. However, DewrapNet is trained once and prediction does not include training. Thus, I am not sure the comparisons are fair. At least, I think the performance issues should be discussed.

Based on the above reason, the practicality of the proposed method is not clear.

**Summary Of The Paper:**

This paper presents document unwrapping based on optimization of neural networks. Four MLPs are defined to model the implicit surface of the target, the colors of the target, the mapping function from 3D points to 2D texture coordinates, and the inverse mapping from 2D to 3D. Similarly to NERF, the target example is approximated by training the MLPs using differentiable rendering.



**Summary Of The Review:**

I think this paper is interesting for an experimental study, where implicit differential rendering is applied for document unwrapping. Although I doubt the practical value of the method.

As an experimental study, the supplemental material seems to include much information about the study that is missing in the paper itself. My concern is that the paper deeply depends on the supplemental document.

Overall, I would like to positively evaluate the trial of implicit differential rendering for document unwrapping, and showing detained results in supplemental document.

---

> ### Author Response · Authors · 2021-11-19
> **Response to Reviewer hfPz**
>
> We thank the reviewer for the comments and suggestions. To make it more comprehensive, we will try to move some of the suggested supplementary contents to the main paper.
>
> **The practicality of the method:**
> We agree that a longer training time per scene is a concern for practical applications. As discussed in section 14 of the supplementary, we believe a faster training alternative of the current method should be considered as future work. Despite the practical limitation, this is the first paper that presents neural rendering-based texture unwarping using multi-view images of a scene. Moreover, our method clearly shows better robustness and generalizability during test time compared to single view methods. We argue that the main advantage of this framework is robust unwarping across different views without any complicated ground-truth data such as the 3D shape or unwarping grid.
>
> **Fairness of Comparison:**
> Because our proposed method is the first learning-based approach that uses multiple images without a ground truth deformation field to improve document unwarping, it is impossible to create a completely apples-to-apples comparison with previous single image-based approaches. However, we provided different evaluation schemes to give the best possible evaluation in terms of fairness. First, we provided a comparison with the average performance of all views from previous art.
> It shows the significant improvement one can achieve by using multi-view images without relying on a synthetic dataset equipped with GT 3D shape and GT unwarping grid. Obtaining the masks and the camera parameters is a fairly simple task, and we can use standard out-of-the-box tools to get these additional inputs.
> Secondly, we also compare the best possible view result that previous work can achieve, assuming there is an oracle view selection algorithm for it.
> Thirdly, we report a new metric suggested by reviewer 1 to address the lack of robustness of LD and MSSIM to subtle global transformations. ~~According to this metric, the proposed method’s mean best score of all views achieves a 17% better score than DewarpNet’s mean best score across all the scenes.~~ It shows our results are much more competitive (9% worse) than comparing LD (22% worse) when compared to DewarpNet's best result across all views. The detailed table can be found in response to reviewer 1. However, we would like to stress the qualitative improvement in this experiment. The figure 5 and 6 in supplementary demonstrates the 5 best unwarping results achieved by DewarpNet in terms of LD score. One can clearly see that visually our results are far better in all cases. We show 4 more scenes demonstrating the same fact in figure 5 of the main submission.

---

### Official Review · Reviewer_QH85 · 2021-11-02

**Correctness:** 3
**Technical Novelty And Significance:** 3
**Empirical Novelty And Significance:** 2
**Recommendation:** 5
**Confidence:** 4

**Main Review:**

**Advancing IDR**

Although the IDR technique that this paper builds upon is able to render textures, these textures must be learned via a MLP trained for appearance information, which is not sufficient to enable texture editing after the fact. In other words, while textures can be rendered an implicit surface via IDR, no explicit surface parameterization is created, so the appearance is fixed. In contrast, the propose method learns the surface parameterization explicitly, enabling the original texture of the implicit surface to be both rendered and subsequently edited. Enabling this subsequent manipulation is a key contribution of this work that suggests real opportunity for further development.

**Simplified Problem**

This paper shows the proposed approach to work quite well for document unwarping. However, document unwarping provides a fairly simplified version of the more general problem of learning explicit texture maps for implicit surfaces.

1. The implicit surface in question (a piece of paper) does not need to be stretched to be unwrapped, so the problem of distortion in the texture space is significantly reduced. Furthermore, no unwarping needs to occur as the texture space actually defines the unwarped document exactly.
2. The implicit surface is already a topological disc, which circumvents the problem of defining the UV space on a more complex surface

While the paper is forthright about these simplifying circumstances, it does somewhat limit the evaluation of the proposed approach in general. It would be helpful for the authors to articulate how this method would generalize beyond documents unwarping and what potential pitfalls may exist in trying to do so.

**Experimental Evaluation**

*Quantitative Results*

The auxiliary metric of OCR performance is apt for document unwarping, and the improvements here are pretty clear (Table 1, supplementary). However, the improvements in Multi-Scale Structural Similarity (MS-SSIM) and Local Distortion (LD) highlighted in Table 1 are not quite as obvious. It seems that while the proposed method outperforms DewarpNet on average, the best view from DewarpNet is  still a little better. Table 1 is missing best view statistics for the proposed method, making it difficult to judge what to make of this evaluation. Additionally, it is not entirely clear what to make of an improvement of 0.05 MS-SSIM. As this is a fairly perceptual measure, it would be helpful for the paper to describe what this improvement means in terms of actual perceptual difference.

Runtime improvement should not be considered "out-of-scope of this paper" (Sec 12, supplementary). This is especially true when the 18-hour training time per scene is significantly higher than the inference time for the primary comparison work, DewarpNet (Sec 14, supplementary). This evaluation belongs in the main paper and needs to be discussed. It is well known that neural volume rendering approaches like IDR are quite slow, and their study is still nascent. However, delegation to the supplementary without any reference in the main paper comes across as evasive. This discussion ought to be present in the main paper to provide a complete and fair evaluation of the proposed technique.

*Qualitative Results*

The qualitative results highlighted in Figure 4 are not particularly discriminative. For example, the DewarpNet result shown in the top row, column (e) seems noticeably sharper than the corresponding result of the proposed approach. Similarly, in the second row, the proposed approach (column (c)) seems to have a less distorted unwarping than DewarpNet (column (b)), but the effects seem pretty marginal in the close up comparison given in column (e). Figure 5 does a better job showing where the proposed approach outperforms DewarpNet.

**Other Notes**

* The OCR evaluation (Sec 10, supplementary) is very relevant for the document unwarping task and is the quantitative result that best distinguishes the proposed approach. The authors ought to consider moving it to the main paper.
* Given the tightly constrained problem, the discussion of limitations (Sec 12, supplementary) and failure cases (Sec 13, supplementary) ought to be part of the main paper.


**Summary Of The Paper:**

**High-Level Overview**

This paper proposes a neural rendering technique to learn how to estimate an implicit surface and its UV-parameterization from a set of 2D images. The proposed approach is applied to the problem of document unwarping, wherein a wrinkled, folded, or bent document (e.g. a piece of paper with text on it) can be digitally flattened with its contents preserved. For good measure, the paper also shows that learning the texture mapping enables the contents of the unwarped document to be edited in the texture space. The proposed approach is evaluated against a prior work, DewarpNet (Das et al., 2019) on the basis of Multi-Scale Structural Similarity (MS-SSIM) and Local Distortion (LD) as well as the auxiliary task of OCR.

**Key Contributions**

* Builds on IDR (Yariv et al., 2020) to learn an explicit bijective texture mapping for a learned implicit surface
* A loss function formulation for the framework
* Application of learned implicit surfaces for document unwarping with reduction in texture distortion and improvements in subsequent OCR metrics

**How It Works**

* Starts with the same problem formulation as IDR: an MLP to model the surface (Eq. 1) and an appearance MLP to model the color (Eq. 2)
* Modifies the appearance MLP to also be a function of texture coordinates (Eq. 3)
* Adds an additional forward and backward MLP to learn the bijective UV mapping
* Jointly trains all MLPs using a pixelwise rendering loss

**Summary Of The Review:**

This paper does a good job demonstrating the applicability of the proposed neural rendering technique to the problem of document unwarping, both in the formulation of the problem (learning the UV grid *is* unwarping the document) as well as in the empirical evaluation (OCR). Design decisions are well-motivated and clearly articulated, and the merits of the proposed technique are backed up by relevant experiments. However, the improvements over prior work seem marginal at best in the key metrics of Multi-Scale Structural Similarity (MS-SSIM) and Local Distortion (LD). Furthermore, the qualitative results do not make the improvements especially obvious. Lastly, the 18 hour training time per scene (Sec 12, supplementary) is pretty prohibitive for the document unwarping application explored by this paper. Given these caveats, it is hard to see how substantially this work might impact the research community.

Additionally, although document unwarping is an interesting application of this neural rendering technique, it is also very niche. Given the potential this paper suggests for learning editable texture-mapped surfaces, it would have been nice to see a more general application (e.g. 3D shapes, objects, or structures) flirted with, even if only as a teaser experiment or discussion of future direction. Despite this wish, it would be inappropriate to knock the submission for not exploring every possible application. However, it does leave an open question as to whether the proposed approach is more generally applicable.

---

> ### Author Response · Authors · 2021-11-19
> **Response to Reviewer QH85**
>
> We thank the reviewer for the detailed comments. The following sections address the major concerns raised by the reviewer:
>
> **General application:**
> We have considered applying our method to fabrics, which are very similar to papers and lead to practical applications of texture editing. However, none of the current 3D garment/fabric datasets can be easily adapted to train the $F_{uv}$ prior. $F_{uv}$ is learned from the mapping of the 3D points to the UV ground-truth data. For more complex UV spaces (e.g., texture atlas), learning the prior may require decomposing the shape to multiple simple UV maps. The proper way to do this is beyond the scope of this paper, however, we believe it’s exciting future work. As importantly, in this paper, we have introduced a number of domain specific strong constraints that suit the rectangular paper shape. These constraints improve empirical results. More general objects will require different constraints e.g., spherical UV domain, local scaling of the conformal map etc.
>
> **Quantitative results:**
> As discussed in section 4.2 paragraph 2, MSSIM and LD numbers are often affected by subtle global transformations, therefore, may not reflect the actual improvement/competitiveness. To this end, we reported a modified version (suggested by reviewer 1) of the local distortion (LD) metric. It first aligns the unwarped image with the GT based on an estimated rigid transformation. ~~According to this metric, the proposed method’s mean best score of all views achieves a 17% better score than DewarpNet’s mean best score across all the scenes.~~ The table can be found in the comment to reviewer 1.
>
> **Qualitative results:**
> We note the few unclear examples reported in figures 4 and 5. To better understand the qualitative improvement, we will move a few high-resolution real examples from the supplementary figures 10, 11, 12, and 13 to the main paper.
>
> **Training time and other limitations:**
> We agree that the current training time makes the proposed approach impractical for real-time application and will move the discussion from the supplementary to the main paper. Extending the proposed method for real-time application will be addressed as future work. Despite these limitations, this is the first paper that presents neural rendering-based texture unwarping using multi-view images of a scene. Moreover, our method clearly shows better robustness and generalizability during test time compared to single view methods. We argue that the main advantage of this framework is robust unwarping across different views without any complicated ground-truth data such as the 3D shape or unwarping grid.

---

### Official Review · Reviewer_K4gj · 2021-11-02

**Correctness:** 3
**Technical Novelty And Significance:** 3
**Empirical Novelty And Significance:** 2
**Recommendation:** 6
**Confidence:** 4

**Main Review:**

Strengths

- The paper is well structured and clearly written which makes it easy to follow and reproduce the method.
- The authors creatively combine ideas from different domains, namely the implicit field representation with a volumetric renderer with the parametric mapping functions and distortion regularizers which allow them to model a real physically valid paper material behaviour.
- The authors experiment on multiple tasks including dewarping, texture editing and OCR and the qualitative results look compelling.
- The Fig. 2 is useful for understanding the pipeline.
- Good ablation of the loss term L_T both quantitatively and qualitatively.

Weaknesses

- Table 1. - It is not clear why the authors did not include the "best view" setting of the proposed method too, it would be a more fair comparison to the "best view" of DewarpNet, as now both methods would be able to use the oracle. Was there a reason for not including this comparison?
- There exist more recent document rectification works which are as recent as DewarpNet or even newer, specificially [1] and [2] (see below), both seem to provide code publicly. Is there a reason why the authors excluded these works for comparison?
- The authors claim that LD and MSSIM are sensitive to perceptually unimportant global transformations such as global translation. Could the authors then first align the GT and the unwarped image by optimizng for the best translation and then evaluate the LD and MSSIM to show that the drop in precision is indeed due to the transformations?
- In Eq. 8, it is not clear how M_p is computed and the description could be more detailed.
- The last paragraph of  page 8: What is "Data 6", does it correspond to the line "Real6" in the Table 1?

Minor problems:
- Fig. 3 is first referenced from the 5th paragraph of the Intro, but at this point the variables used in the caption are not defined and it is hard to interpret the figure. I would suggest either referencing the figure later from section 3.2, or giving more comprehensive self-contained caption.
- Section 3.4 Training Details: In the last sentence, the authors probably meant \delta_2, \delta_3, \delta_4 instead of \delta_1, \delta_2, \delta_3.
- Typos:
	- Section 3.2, paragraph Deformation constraints for F_z: "conformal map allow" -> "conformal map allows"
	- Fig. 5: "best" -> "the best"

A question for the authors:
Is L_{uv} of Eq. 10 really needed? The loss terms L_E, L_F and L_G should exactly such behaviour where the mapping does not introduce strong local distortions. The authors explain that if L_{uv} is not employed, the effective learned UV space does not span the full rectangle [0, 1]^{2}. However, at the same time, the authors show that the distortion regularizers L_E, L_F, L_G are only turned on later during the training. Could the authors comment on what happens if the the terms L_E, L_F, L_G are turned on from the beginning and the term L_{uv} is not used at all?

References
[1] X. Li et al. Document Rectification and Illumination Correction using Patch-based CNN. TOG 2019.
[2] X. Liu et al. Geometric rectification of document images using adversarial gated unwarping network. Pattern Recognition 2020.

**Summary Of The Paper:**

The authors present a method to jointly learn a shape and UV parameterization of a creased document observed from multiple views. They use an SDF based implicit shape representation in combination with a recent neural volumetric renderer IDR, and optimize the energy composed of the main photometric term (a discrepancy between the observed and rendered image) and regularization terms enforcing well behaved UV space and a fixed scale conformal mapping which corresponds to a behaviour of a real paper material. Since the implicit field representation does not explicitly model a surface, the authors cannot directly optimize the 3D->2D mapping using standard graphics techniques and rather adapt the IDR renderer to be conditioned on the locations sampled from the UV space and train it jointly with the MLPs defined for the bijective mapping 2D->3D and 3D->2D. The authors show that they can get better results both quantitatively and qualitatively when compared to one of the SotA works DewarpNet.

**Summary Of The Review:**

The paper introduces a creative way to combine an implicit shape representation with parametric functions and a neural volumetric renderer to deal with document unwarping. The authors further employ deformation regularizers which reflect the real behaviour of a material on which the documents are usually printed. The paper is written in a clear way, I believe it should be easy to reproduce, and the qualitative results seem compelling. I see a couple minor problems, namely (i) exclusion of two seemingly relevant approaches for comparison (see above), (ii) a little confusion about the column "best view" in the table not being used for the proposed method too and (iii) unsupported claim about the sensitivity of LD and MSSIM and I would like the authors to comment on these. In general, I find this submission to be compelling overall and upon clarification of some of the issues mentioned above I lean towards acceptance.

---

> ### Author Response · Authors · 2021-11-19
> **Response to Reviewer K4gj**
>
> We thank the reviewer for the insightful review.  Following sections address the major concerns:
>
> **Exclusion of [1] Li et al. and [2] Liu et al.:**
> We have tried to compare with [1], [2] and found that DewarpNet (DN) is significantly better. A direct quantitative comparison of [1], [2], and DN can be found in the Piecewise Unwarping paper [3]. Therefore, we chose DewarpNet to provide the strongest possible baseline. Moreover, [2] only provides the synthetic data generation code; neither model nor training code is available.
>
> **Sensitivity of LD and MSSIM:**
> As suggested by the reviewer, we compare the proposed method and DN with a modified version of LD. We first estimate a rigid transformation (scale and translation) and align the unwarped and GT image before calculating the local distortion. This new score is reported in the table below and includes : (a-b) All views (X) refers to the mean result of all views by method X; (c) DewarpNet's Best View (Proposed) refers to the results of the proposed method on the views which have the best score with DN; (d-e) Best View (X) refers to the best score obtained by X across all views; (f) Proposed's best view (DewarpNet) refers to results of DN on the views which have the best score with the proposed method. As per this metric, the proposed method's mean score of all views improves ~~DN's best score by 4% and improves~~ DN's mean score of all views by 43%. ~~In this table, we also report the best score of all views across all the scenes, for which the proposed method achieves a 17% better score than DN's best score views.~~ With this metric the best view results are much more competitive than comparing LD. However, we uld like to stress proposed's qualitative improvement in comparison with the best result of DewarpNet. The figure 5 and 6 in the supplementary demonstrate the 5 best unwarping results achieved by DewarpNet. One can clearly see that visually our results are far better in all cases. We show 4 more scenes demonstrating the same fact in figure 5 of the main submission.
>
> ~~| Scene | (a) All views (Proposed) | (b) All views (DewarpNet) | (c) DewarpNet's Best View (Proposed) | (d) Best View (DewarpNet) | (e) Best View (Proposed) | (f) Proposed's Best View (DewarpNet) |
> |:---:|:---:|:---:|:---:|:---:|:---:|:---:|
> | Synth 3 | 0.047 | 0.095 | 0.466 | 0.468 | **0.042** | 0.116 |
> | Mean | **0.161** | 0.283 | 0.185 | 0.168 | **0.138** | 0.221 |~~
>
> Updated after Reviewer QH85 noted the discrepancy with Synth 3 results:
>
> | Scene | All views (Proposed) | All views (DewarpNet) | DewarpNet's Best View (Proposed) | Best View (DewarpNet) | Best View (Proposed) | Proposed's Best View (DewarpNet) |
> |:---:|:---:|:---:|:---:|:---:|:---:|:---:|
> | Synth 1 | 0.134 | 0.291 | 0.131 | **0.113** | 0.127 | 0.209 |
> | Synth 2 | 0.071 | 0.151 | 0.067 | 0.079 | **0.061** | 0.147 |
> | Synth 3 | 0.047 | 0.095 | 0.047 | 0.047 | **0.042** | 0.116 |
> | Synth 4 | 0.271 | 0.425 | 0.193 | 0.196 | **0.189** | 0.238 |
> | Synth 5 | 0.362 | 0.484 | 0.311 | **0.204** | 0.296 | 0.354 |
> | Synth 6 | 0.236 | 0.326 | 0.234 | **0.193** | 0.227 | 0.294 |
> | Synth 7 | 0.123 | 0.258 | 0.106 | **0.100** | 0.105 | 0.125 |
> | Synth 8 | 0.149 | 0.222 | 0.139 | **0.132** | 0.136 | 0.198 |
> | Synth 9 | 0.113 | 0.269 | 0.105 | **0.087** | 0.101 | 0.216 |
> | Synth 10 | 0.103 | 0.306 | 0.096 | 0.111 | **0.096** | 0.310 |
> | Mean | 0.161 | 0.283 | 0.143  | **0.126** | 0.138 | 0.221 |
>
> By comparing the mean of all views (All views (Proposed)) and best of all views (Best View (Proposed)), we can see the best result is very close to the mean, therefore explicitly showing our methods' robustness across different views.
>
> **Utility of $L_{uv}$**:
> We have noticed that introducing $L_F$, $L_E$, $L_G$ from the start causes instability in training. Initially, while the surface is being optimized, $F_{uv}$ prediction changes. Consequently, $F_{z}$ also changes which can result in unstable values of L_F, L_E, L_G. We found that a stable and time-efficient training strategy for the proposed framework is to initially train the surface and gradually introduce the UV training and geometric regularizations. Moreover, introducing $L_F$, $L_E$, $L_G$ from the start of the training is time inefficient because these terms are explicitly calculated using the Jacobian of the learned surface. On the contrary, $L_{uv}$ provides a faster, simplistic solution to constrain the UV space.
>
> References:
> [1] Li et al. Document Rectification and Illumination Correction using Patch-based CNN. TOG 2019.
> [2] Liu et al. Geometric rectification of document images using adversarial gated unwarping network. Pattern Recognition 2020.
> [3] Das et al. End-to-End Piece-Wise Unwarping of Document Images. ICCV 2021.

---

> > ### Comment · Reviewer_QH85 · 2021-11-22
> > **What's up with Synth 3?**
> >
> > Looking at these new results, it seems Synth 3 is the reason why the proposed solution outperforms DewarpNet. Synth 3 is the only case where there's a significant difference between the proposed approach and DewarpNet. If you remove Synth 3, the mean for DewarpNet is **0.135** and the mean for the proposed approach is **0.149**.  That is, the proposed method actually *underperforms* its baseline by quite a bit. The authors' point about lower variance in performance is well taken, but it is also to be expected given that DewarpNet aims to be a general solution, while neural rendering fits to a given input. I'm hesitant to accept these results as particularly conclusive support for the proposed approach.
> >
> > This is a nice work and an interesting application of neural rendering, but the quantitative results (and the qualitative ones provided in the paper) don't really justify the longer training time and lack of generalization.

---

> > > ### Author Response · Authors · 2021-11-23
> > > **Typo in Synth 3 results**
> > >
> > > Thank you for pointing out the discrepancy with Synth 3. We rechecked the values and realized there was a typo in row synth 3 column c and column d. 0.467 should be 0.0467 and 0.468 should be 0.0468. Following is the updated table:
> > >
> > > | Scene | All views (Proposed) | All views (DewarpNet) | DewarpNet's Best View (Proposed) | Best View (DewarpNet) | Best View (Proposed) | Proposed's Best View (DewarpNet) |
> > > |:---:|:---:|:---:|:---:|:---:|:---:|:---:|
> > > | Synth 1 | 0.134 | 0.291 | 0.131 | **0.113** | 0.127 | 0.209 |
> > > | Synth 2 | 0.071 | 0.151 | 0.067 | 0.079 | **0.061** | 0.147 |
> > > | Synth 3 | 0.047 | 0.095 | 0.047 | 0.047 | **0.042** | 0.116 |
> > > | Synth 4 | 0.271 | 0.425 | 0.193 | 0.196 | **0.189** | 0.238 |
> > > | Synth 5 | 0.362 | 0.484 | 0.311 | **0.204** | 0.296 | 0.354 |
> > > | Synth 6 | 0.236 | 0.326 | 0.234 | **0.193** | 0.227 | 0.294 |
> > > | Synth 7 | 0.123 | 0.258 | 0.106 | **0.100** | 0.105 | 0.125 |
> > > | Synth 8 | 0.149 | 0.222 | 0.139 | **0.132** | 0.136 | 0.198 |
> > > | Synth 9 | 0.113 | 0.269 | 0.105 | **0.087** | 0.101 | 0.216 |
> > > | Synth 10 | 0.103 | 0.306 | 0.096 | 0.111 | **0.096** | 0.310 |
> > > | Mean | 0.161 | 0.283 | 0.143  | **0.126** | 0.138 | 0.221 |
> > >
> > > As per these changes, we must revise our observations. The mean of the best score by DewarpNet across all views provides the best overall quantitative result. Note that, the proposed method's best score across all views is competitive (relatively only 9% worse) to DewarpNet's best. However, we would stress that the qualitative improvement in this experiment is substantial. The figure 5 and 6 in the supplementary demonstrate the 5 best unwarping results achieved by DewarpNet. One can clearly see that visually our results are far better in all cases. We show 4 more scenes demonstrating the same fact in figure 5 of the main submission which compares DewarpNet's best and corresponding proposed unwarping. In summary, our quantitative results are comparable and qualitative results are much better than previous art.

---

> > > > ### Comment · Reviewer_QH85 · 2021-11-23
> > > > **Authors might consider revising and resubmitting this work**
> > > >
> > > > I stated in my main review and reiterated above that this is a nice paper, and I stand by that comment. There's a lot to like about the formulation of the proposed method and the application space. However, there are too many concerns with the submission in its current condition. I think the authors have worthwhile research here, but based on my read, the other reviews, and the updates above, I recommend that the authors consider re-evaluating their results, re-framing their argument, and re-submitting this paper to a future venue. There are some good, actionable steps that have been suggested in these reviews that can help take this paper to the next level. If the authors take this feedback to heart, I think this submission would have a good chance at a future venue.
> > > >
> > > > **Note:** The authors also deserve credit for acknowledging the error in their reported results. It would have been easy to ignore the question. Instead, they chose the honest route, despite it undermining their original observations.

---

> > ### Comment · Reviewer_K4gj · 2021-11-30
> > **Major concerns clarified**
> >
> > The authors have clarified my major concerns, namely (i) the reason for the excluding the comparison to [1], [2], (ii) the exclusion of "Best View" setting for DewarpNet and (iii) evaluating LD and SSIM after rigidly prealigning the prediction with the GT. I provide more detail below.
> >
> > (i) The authors mention the work [3] which itself compares [1] and [2] to DewarpNet and, indeed, shows that DewarpNet outperforms both [1] and [2]. The authors could thus possibly compare the proposed work to [3] as well, but it is an ICCV'21 paper which is too recent to be considered for comparison within ICLR'22. Therefore, I believe the selected baselines are sufficient.
> >
> > (ii), (iii) The authors presented a new table where they report the values on the LD metrics after rigidly prealigning the predictions and GT. The results indicate that the authors yield higher mean accuracy and lower "best view" accuracy as compared to DewarpNet. While the "best view" setting is quantitatively not compelling, I would argue that the results are, in general, satisfactory, as the mean accuracy seems to be more meaningful for practical scenarios (e.g. taking pictures with a smartphone from a random viewpoint), where the accuracy averaged over various views seems more important. That being said, I agree with the concerns of R QH85 and R hfPz about the excessive runtime of 18 h per scene. On the other hand, being a research work, the method can potentially be expected to be sped up with future research work.
> >
> > I agree with R QH85 that the OCR experiments results are quite essential and should be moved from the supplementary to the main paper.
> >
> > Overall, my main concerns have been addressed by the authors. I share some of the concerns of the other reviewers, namely the fact that for some of the qualitative results (e.g. in Fig. 4) it is not immediately visually clear that the proposed method brings a significant improvement. However, some of the other results, such as the OCR, are compelling. Furthermore, the long runtime might be concerning, but in my opinion the technical contribution, namely the capability of the proposed method to explicitly model the surface which allows e.g. for retexturing, outweighs such problem. In conclusion, I am leaning towards acceptance.

---

### Decision · Program_Chairs · 2022-01-20

**Decision:**

Reject

**Comment:**

This paper proposes an architecture for learned surface parameterization, with application to image unwarping, which can be coupled with differentiable rendering, multi-view data, and other modern objective terms.  The shape of the document is parameterized using an SDF technique, coupled with neural rendering and objective terms inspired by classical geometry processing.  This machinery is quite "heavy," leading to slow training times.

As pointed out by reviewer QH85, there were some experimental discrepancies---rightfully acknowledged by the authors---which make comparisons to DewarpNet less favorable for the new method, at least from a quantitative perspective.  Visual inspection makes the comparison more favorable, although it would be preferable for the quantitative quality metrics and qualitative examples to align.

Runtime measurements here are also not favorable and severely limit applicability of this technique in real-world scenarios, as pointed out by reviewers hfPz and QH85.

While the mistaken quantitative results are forgivable, the AC agrees that the scope of this work is quite narrow; it is not clear where this architecture would be applied relative to the motivating application.